# The CDK4/6 Inhibitor Palbociclib Inhibits Estrogen-Positive and Triple Negative Breast Cancer Bone Metastasis In Vivo

**DOI:** 10.3390/cancers15082211

**Published:** 2023-04-08

**Authors:** Lubaid Saleh, Penelope D. Ottewell, Janet E. Brown, Steve L. Wood, Nichola J. Brown, Caroline Wilson, Catherine Park, Simak Ali, Ingunn Holen

**Affiliations:** 1Mellanby Centre for Musculoskeletal Research, Department of Oncology and Metabolism, University of Sheffield, Sheffield S10 2RX, UK; lubaid.saleh@sheffield.ac.uk (L.S.);; 2Weston Park Hospital, Whitham Road, Sheffield S10 2SJ, UK; 3Department of Surgery and Cancer, Imperial College London, Hammersmith Hospital Campus, London W12 0NN, UK

**Keywords:** triple negative breast cancer, metastasis, bone, cdk, cdk inhibitors, palbociclib

## Abstract

**Simple Summary:**

When breast cancer (BC) spreads to the skeleton it is no longer curable; new treatments are needed in this setting. When given daily, the CDK4/6 inhibitor palbociclib significantly impedes tumour growth in murine models of both estrogen receptor positive and triple negative bone metastatic BC. When a treatment break was introduced, mimicking the clinical setting, tumour growth resumed and continued even in the presence of further cycles of palbociclib. In combination treatment with bisphosphonate zoledronic acid, or a CDK7 inhibitor, palbociclib was insufficient in preventing tumour growth. This suggests that tumour cells become insensitive to palbociclib after a treatment break. To explore possible underlying reasons for this, we harvested palbociclib-sensitive and -insensitive tumour cells from bone and found differences in the levels of key proteins that palbociclib affects. We provide the first demonstration that palbociclib is effective at reducing breast tumour growth in bone, if given daily.

**Abstract:**

CDK 4/6 inhibitors have demonstrated significant improved survival for patients with estrogen receptor (ER) positive breast cancer (BC). However, the ability of these promising agents to inhibit bone metastasis from either ER+ve or triple negative BC (TNBC) remains to be established. We therefore investigated the effects of the CDK 4/6 inhibitor, palbociclib, using in vivo models of breast cancer bone metastasis. In an ER+ve T47D model of spontaneous breast cancer metastasis from the mammary fat pad to bone, primary tumour growth and the number of hind limb skeletal tumours were significantly lower in palbociclib treated animals compared to vehicle controls. In the TNBC MDA-MB-231 model of metastatic outgrowth in bone (intracardiac route), continuous palbociclib treatment significantly inhibited tumour growth in bone compared to vehicle. When a 7-day break was introduced after 28 days (mimicking the clinical schedule), tumour growth resumed and was not inhibited by a second cycle of palbociclib, either alone or when combined with the bone-targeted agent, zoledronic acid (Zol), or a CDK7 inhibitor. Downstream phosphoprotein analysis of the MAPK pathway identified a number of phosphoproteins, such as p38, that may contribute to drug-insensitive tumour growth. These data encourage further investigation of targeting alternative pathways in CDK 4/6-insensitive tumour growth.

## 1. Introduction

Affecting 2.26 million women each year, breast cancer (BC) is the most common cancer worldwide [1]. It is estimated that BC resulted in over 600,000 deaths in 2018 with 11,500 in the UK. ER+ve BC accounts for 70% of cases whereas triple negative BCs (TNBC) account for approximately 20% and are considered as the most aggressive subtype of BC with the poorest prognosis [2]. TNBCs lack estrogen (ER)/progesterone (PR) as well as HER-2 receptor expression, and this absence of druggable targets mean that TNBC remains a challenging disease to treat. Furthermore, treatment for TNBC becomes palliative in the metastatic setting with a median overall survival (OS) of 13.3 months [3].

Approximately 80% of patients with advanced BC will have skeletal involvement [4], associated with clinical symptoms such as bone pain, spinal cord compression and fractures [5]. Metastasis to the bone is considered an incurable disease and treatments are palliative with patients having a median survival of 24–54 months [6]. Systemic therapy still proves to be the standard treatment in the management of all metastatic BC subtypes to the bone [7]. In bone metastasis derived from ER+ve tumours, hormone therapy, radiotherapy, chemotherapy, a combination of both and weekly intravenous bisphosphonate therapy (such as zoledronic acid) are commonly used [8,9]. However, options remain limited for patients with TNBC bone metastasis where anthracycline- or taxane-based therapies are the standard of care treatments [10,11].

The lack of effective therapies in metastatic TNBC has driven the search for alternative targetable pathways. One such pathway is the cyclin-dependent kinase (CDK)–cyclin–retinoblastoma (Rb) interaction. The CDKs are a family of serine/threonine kinases that are key regulators of the cell cycle. Dysregulation of normal cell cycle control results in abnormal proliferation, a characteristic and hallmark of cancer [12]. CDK4/6 (a subgroup of CDKs) has been shown to play a role in BC pathogenesis with the overexpression of cyclin D1 found in 50% of all BC cases leading to aberrant phosphorylation of Rb and irreversible cell cycle progression [13]. The development of CDK inhibitors over the past 25 years has therefore resulted in the discovery of the CDK4/6 specific inhibitor, palbociclib [14]. Early trials have demonstrated the safety of palbociclib with improved median progression free survival (PFS) and clinical benefit in ER+ve BC when administered in a 3 week-on, 1 week-off cycle [15,16,17,18,19]. Currently palbociclib is predominantly used for endocrine therapy-resistant ER+ve breast cancers, demonstrating significant improvement in patient PFS and OS when combined with endocrine therapy [20]. Furthermore, palbociclib has been shown to improve PFS and OS in patients with metastatic ER+ve/PR+ve BC in combination with endocrine therapy [17,21]. Thus, less work has been carried out to investigate the effectiveness to CDK4/6 inhibitors in TNBC.

TNBC tumours have been shown to demonstrate sensitivity to palbociclib in vitro and in vivo [22,23]. Moreover, it is suggested that palbociclib can sensitise TNBC cells to chemotherapeutic agents [24,25,26]. Current clinical trials are assessing the effects of palbociclib in combination with chemotherapeutic agents in metastatic TNBC patients (ClinicalTrials.gov identifier: NCT02605486) [27,28]. CDK4/6 inhibitors have now become standard practice and are being extended to the adjuvant setting for the treatment of ER+ve metastatic disease [29].

Here, we used an ER+ve model of spontaneous breast cancer metastasis from the mammary fat pad to bone and a TNBC model of metastatic outgrowth in bone (intracardiac route), to provide the first demonstration that palbociclib inhibits both ER+ve and TNBC growth in murine bone and to establish potential mechanisms that promote CDK4/6 targeted drug insensitivity following treatment breaks.

## 2. Materials and Methods

### 2.1. Animals

The TNBC bone metastasis model was generated using 6-week-old female BALB/c Nude mice (Charles River, Kent, UK). Fourteen-week-old female BALB/c mice were used for the ER+ BC model. All mice were maintained on a 12 h:12 h light/dark cycle with free access to food and water. 12 mg/mL 17β-estradiol (Sigma-Aldrich, St. Louis, MO, USA) was added to the drinking water was used to supplement ER+ BC growth. Mice that did not develop skeletal tumours due to unsuccessful injection of MDA-MB-231Luc2+GFP+ into the left ventricle of the heart were removed from the study. Mice underwent stratified randomization in the ER+ve study 4 weeks after tumour cell injection.

### 2.2. Cell Lines and Culture

The triple negative human breast cancer MDA-MB-231Luc2+GFP+ and the ER+ve T47DLuc2+GFP+ cell lines were stably transfected with luciferase and GFP [30] and cultured in RPMI + 10% FCS (Gibco, Invitrogen, Paisly, UK). ER+ve (MCF7) cells were cultured in RPMI + 10% FCS and DMEM + 10% FCS (Gibco, Invitrogen, Paisly, UK), respectively. All cells were cultured in a humidified incubator under 5% CO_2_. For tumour cell injection, 5 × 10^4^ cells were resuspended in 100 µL PBS and kept on ice.

### 2.3. Drugs

Palbociclib (PD-0332991, Pfizer) was dissolved in sodium lactate (pH4) solution (Sigma) for in vitro and in vivo studies. The CDK7 inhibitor ICEC0942 was obtained from Professor Ali [31] and dissolved in PBS. Zoledronic acid (Zol) [(1-hydroxy-2-(1H-imidazol-1-yl) ethylene) bisphosphonic acid] was obtained from Novartis Pharma AG (Basel, Switzerland) and dissolved in PBS. CDK inhibitors were administered by oral gavage and Zol via i.p. injection.

### 2.4. Dose Response Assay

MDA-MB-231Luc2+GFP+ were seeded in triplicate at 3000 cells per well in a 96 well plate. At 24 h after plating, cells were treated with 1 mM palbociclib and 10-fold dilutions over six concentrations to generate a dose response curve. Control cells without drug, vehicle (sodium lactate) and blank (no cells) wells were also plated. Cells were incubated with the drug for 72 h, after which the MTT assay (Sigma) was conducted to measure cell number. The SpectreMax M5e multiwell plate reader (Molecular Devices, LLC, Sunnyvale CA, USA) was used at 570 nm absorbance.

### 2.5. Western Blot Analysis

A total of 7 × 10^5^ MDA-MB-231, MCF7 and T47D cells were plated in a T25 flask and treated with the calculated respective IC_50_ values of palbociclib (or vehicle) 24 h later. Cells were incubated with drug or vehicle for 72 h and harvested by washing in ice-cold PBS and scraping. A lysis buffer consisting of 1% protease inhibitor cocktail (Sigma) and RIPA buffer (Sigma) was used to lyse the cells at 4 °C. Insoluble material was removed by centrifugation at 12,000 rpm for 20 min. Protein quantification was conducted by NanoDrop^TM^ 2000 spectrophotometer (Thermo Scientific, Wilmington, DE, USA). Proteins were resolved by SDS-PAGE electrophoresis in a 4–20% Tris-Glycine precast gel (Bio-Rad, CA, USA) and transferred to a PVDF membrane (Merck Millipore Billerica, USA). Total Rb and pRb protein expression was detected using Rb (D20) and phospho-Rb (Ser780) rabbit monoclonal antibodies, respectively (Cell Signalling Technologies, Beverly, MA, USA). Blots were incubated with polyclonal goat-anti-rabbit IgG horseradish peroxidase conjugate (DAKO) and developed with West Femto Maximum Sensitivity Substrate (ThermoFisher Scientific). Chemifluorescence was used to detect phosphorylated proteins using the Bio-Rad Gel Doc (Image Lab version 6.0.1).

### 2.6. Cell Cycle Analysis

To measure cell cycle arrest in response to palbociclib, MDA-MB-231Luc2+GFP+ cells were plated at a density of 4 × 10^4^ cells per well in a 12 well plate. Cells were treated with palbociclib (0.85 μM) or vehicle for 72 h. Cells were then harvested and permeabilised with ice-cold ethanol (70%) and incubated overnight at 4 °C, washed with PBS and centrifuged at 2500 RPM for 5 min. The pellet was resuspended in 50 μg/mL propidium iodide and 10 μg ribonuclease (Sigma). The solution was incubated for 30 min in the dark and then analysed by flow cytometry using the BD^TM^ LSR II (Becton Dickinson, San Jose, CA, USA). To study the effects of palbociclib after drug removal, cells were plated at the above densities and treated with palbociclib as before for 72 h. The drug was then removed, and the cells were washed with PBS, then cultured in complete medium. Cells were harvested following the above method after 24, 48 and 72 h and analysed by flow cytometry.

### 2.7. Flow Cytometry Analysis for Detection of pRb

MDA-MB-231 cells were plated at a density of 4 × 10^4^ cells per well in a 12 well plate and treated with palbociclib (0.85 μM) or vehicle for 72 h. Cells were harvested by trypsinisation and centrifuged at 2500 RPM for 5 min. After removal of the supernatant, the pellet was resuspended in 100 μL 4% formaldehyde and incubated for 15 min at ambient temperature. Cells were washed with excess PBS and centrifugation. Cells were then permeabilised by adding ice-cold methanol (100%) while gently vortexing and placed on ice for 10 min. After washing with excess PBS, the cells were stained with PE-conjugated Phospho-Rb Rabbit monoclonal antibody, 1:50 dilution (Cell Signalling Technologies), for 1 h in the dark at ambient temperature. Cells were washed with PBS and resuspended in a final volume of 200–500 μL PBS and analysed by flow cytometry using the BD^TM^ LSR II (Becton Dickinson, San Jose, CA, USA).

### 2.8. In Vivo Studies

It has been demonstrated that young (~6 weeks old) BALB/c nude mice with high bone turnover rates develop osteolytic bone metastases more readily following intra-cardiac injection of human breast cancer cells, compared to older mice (aged 12+ weeks) with a more mature skeleton and lower bone turnover [32,33]. Therefore, we used 6-week-old female BALB/c nude mice (*n* = 8/group) and injected 5 × 10^4^ MDA-MB-231Luc2+GFP+ cells via the intracardiac route to generate the TNBC model. The procedure was carried out as described in [33]. Although not including dissemination from a primary tumour, this model does capture the early stages of development of bone metastases, with single cells colonising skeletal sites and subsequently developing into colonies progressing to overt metastases over the following 3–6 weeks. This model is commonly used to investigate effects of therapeutic approaches to bone metastasis [34,35], resulting in skeletal tumour growth in 90% of animals following successful injection. IVIS imaging was carried out 24–48 h after tumour cell injection and any animal displaying a signal in the chest cavity, indicative of a misplaced injection, was removed from the experiment (5% of the total mice in the current study).

Palbociclib treatment schedules: In the TNBC model we explored a number of different treatment schedules, comparing the effects of palbociclib administered 5 days/week vs. 7 days/week for 4 weeks, as well as the effect of starting treatment at different time points (7 days vs. 3 days following tumour cell injection) and whether the addition of an anti-resorptive agent would further reduce tumour growth in bone. Following tumour cell injection, initial experiments were carried out where mice were treated with 100 mg/kg palbociclib (100 μL by oral gavage), or vehicle (100 μL Na lactate, pH4), 7 days following tumour cell injection. Palbociclib was administered for 5 consecutive days followed by a 2-day treatment break. This continued for 4 cycles. In subsequent experiments mice were treated with 100 mg/kg palbociclib (or vehicle) for 28 days consecutively with treatment commencing 3 days after tumour cell intracardiac injection (*n* = 8/group). A third group (*n* = 8) of mice were treated with daily 100 mg/kg palbociclib in combination with once weekly administration of 100 μg/kg of Zol once a week by intraperitoneal injection for the duration of the experiment.

To study the effects of a second cycle of palbociclib, mice (*n* = 8) underwent intracardiac injection of 5 × 10^4^ MDA-MB-231Luc2+GFP+ cells and then commenced palbociclib (100 mg/kg) treatment after 3 days. Administration of palbociclib continued for 28 consecutive days, followed by a 7-day break and a second 28-day treatment period. The CDK7 inhibitor, ICEC0942 (100 mg/kg), was also administered orally combined with palbociclib during the second treatment cycle.

The ER+ T47D (Luc2+GFP+) cell line was used to generate the ER+ BC mouse model as previously described by us [36]. Fourteen-week-old BALB/c female mice were supplemented with 12 mg/mL 17β-estradiol [36] via their drinking water 2 days prior to hind mammary fat bilateral tumour cell injection (1 × 10^6^ cells) and continued to be supplemented with 17β-estradiol throughout the duration of the study. Daily palbociclib treatment commenced 4 weeks following tumour cell injection, by which time bone metastases were evident, and continued for 4 weeks (Appendix A). Tumour progression was monitored in all studies by weekly in vivo live imaging (IVIS) and mice were culled at the respective endpoints with hind limbs subjected to ex vivo imaging to confirm tumour growth. Primary tumour growth was measured twice per week using callipers.

### 2.9. Fluorescent Activated Cell Sorting of TNBC Cells and Human MAPK Downstream Analysis

MDA-MB-231Luc2+GFP+ were harvested from the hind limbs of tumour-positive mice. Tibia and femurs were dissected, centrifuged to harvest bone marrow and crushed using a pestle and mortar. The single cell suspension of bone and bone marrow underwent red blood lysis using ammonium chloride (STEM cell technologies, Vancouver, BC, Canada). Lysed samples were stained with TOPRO-3 (Thermo Fisher, Waltham, MA, USA) to determine live cells. GFP+ tumour cells were identified using the FACSMelody (BD).

Once GFP+ tumour cells were isolated from vehicle and treated animals, cells were lysed according to the Human MAPK phosphokinase array instructions (Abcam, Cambridge, UK). Protein concentrations were measured using a BCA assay and protein blots conducted as per supplier’s instructions (abcam- ab211061). Blots were imaged using the Bio-Rad Gel Doc system and software (Image Lab version 6.0.1) to determine band intensities.

### 2.10. Statistical Analysis

Statistical analysis was conducted on GraphPad Prism version 9.5.0. Student’s *t*-test was conducted to compare results between two groups. Additional details are provided in the figure legends.

## 3. Results

### 3.1. Palbociclib Inhibits the Phosphorylation of Rb and Promotes Cell Cycle Arrest in Triple Negative Breast Cancer Cell Lines

Palbociclib has been demonstrated to significantly improve progression free survival with limited adverse effects in clinical studies of ER+ BC; however, the effects of palbociclib on bone metastasis in TNBC remain to be determined. We therefore investigated the effects of palbociclib using in vivo models of bone metastasis. To confirm that TNBC cells are sensitive to palbociclib in vitro, we demonstrated a dose-dependent decrease with an MTT assay resulting in an IC_50_ of 0.85 µM in our MDA-MB-231Luc2+GFP+ cells used in subsequent in vivo studies (Figure 1A). We further demonstrated the inhibitory effects of palbociclib in alternative ER+ve (MCF7 and T47D-Luc2+GFP+) cell lines in vitro, hence the inhibitory effects of palbociclib were not limited to the two lines used in subsequent in vivo studies (Appendix A).

Inhibiting CDK4/6 activity prevents the phosphorylation of Rb (pRb), which is required for the transition through the G1 cell cycle checkpoint. To determine the downstream effects of palbociclib, Western blot analysis was undertaken, which showed a 52% reduction in pRb protein levels in response to palbociclib (0.85 µM) treatment of the MDA-MB-231Luc2+GFP+ cells in vitro (Figure 1B). This was also demonstrated by flow cytometry (Figure 1C) where palbociclib (0.85 µM) resulted in a three-fold decrease in pRb compared to vehicle treated TNBC cells (*p* < 0.001).

The pRb protein initiates the transition from G0/G1 of the cell cycle to the S-phase; we therefore assessed the cell cycle phases in TNBC cells in response to palbociclib treatment. After a 72 h exposure to 0.85 µM palbociclib, a significantly larger proportion of MDA-MB-231 cells remain in G0/G1 compared to the vehicle control (palbociclib: 89.4%, vehicle: 48.8% *p* < 0.0001) (Figure 1D). We also investigated the duration of the effects of palbociclib on cell cycle distribution following drug removal. MDA-MB-231 cells were treated with 0.85 µM palbociclib (or vehicle) for 72 h, washed and incubated in drug-free medium for an additional 72 h. Cell cycle analysis was conducted 24, 48 and 72 h following drug removal. A full 24 h after drug removal, 56.5% of palbociclib pre-treated cells remained in the G0/G1 phase, compared to 50.9% in the vehicle treated control group (Figure 1E). This proportion significantly increased to 84.9% at 48 h (*p* < 0.005) and remained at this level at 72 h (77.5%) (vehicle 48 h: 35.2%, 72 h: 35.6%). These data confirm that palbociclib causes cell cycle arrest by inhibiting phosphorylation of Rb and that this inhibitory effect remains evident 72 h after removal in the MDA-MB-231Luc2+GFP+ TNBC cell line.

### 3.2. Palbociclib Inhibits Bone Metastasis in ER+ve BC

Palbociclib is effective in the clinical treatment of ER+ BC, with overall survival being significantly improved in patients receiving a combination of palbociclib and endocrine therapy compared to endocrine therapy alone [37]. Furthermore, clinical trials using palbociclib have included a small number of patients with bone-only metastases who demonstrated improved progression free survival (PFS); however, no specific analysis of progression in bone was performed [38]. To determine if CDK4/6 inhibition could reduce skeletal tumour growth, we first investigated the effects of palbociclib on the early stages (defined as growth of disseminated tumour cells in bone) of bone metastasis from ER+ primary tumours in vivo. To do this, we bilaterally injected the ER+ T47D (luc2+GFP+) cells into the hind mammary fat pads of 14-week-old mice and allowed the primary tumours to establish for 4 weeks, at which time tumour growth in bone was detectable. We then proceeded with daily treatment of vehicle or palbociclib (100 mg/kg) for a further 4 weeks (Figure 2A). As expected, we observed consistent primary tumour growth in vehicle treated animals. However, in palbociclib treated animals, primary tumour growth was impeded after 4 weeks of treatment. Although palbociclib significantly inhibited primary tumour growth compared to control (Figure 2B,C, *p* < 0.05), we did not observe statistically significant tumour regression in palbociclib treated animals. In addition, significantly fewer skeletal tumours were observed in palbociclib treated mice compared to vehicle treated mice (Figure 2D, *p* < 0.05). After the 4-week treatment period, hind limbs were harvested (all muscle and associated soft tissues removed) to confirm tumour presence in bone by ex vivo imaging (Figure 2E). We found that significantly fewer animals exhibited skeletal tumours in the palbociclib group compared to the vehicle group (*p* < 0.05). These results demonstrate that in our in vivo model of ER+ breast cancer, palbociclib inhibits both primary tumour growth and development of skeletal metastases.

### 3.3. Intermittent Palbociclib Treatments Does Not Inhibit the Progression of Metastatic BC Growth in Bone

Having demonstrated the inhibitory effects of palbociclib on ER+ve tumour growth, including bone metastases, we next evaluated the effects in an in vivo model of TNBC bone metastasis. A total of 5 × 10^4^ MDA-MB-231Luc2+GFP+ cells were injected via the intracardiac route (i.c.), a model shown to result in skeletal tumour growth (mainly in the long bones of the hind limbs) within 3–4 weeks [30]. In order to allow tumour cells to home to the bone, palbociclib treatment (100 mg/kg) commenced 7 days after tumour cell injection. To establish tolerability, treatment was initially administered in a 5-day-on/2-day-off cycle for 4 weeks (Figure 3A). In vivo live imaging was used to monitor tumour progression throughout the study (Figure 3B). We found that the 5-day-on/2-day-off palbociclib schedule caused a significant reduction in skeletal tumour burden at every timepoint measured, demonstrating that CDK4/6 inhibition can impede TNBC growth in bone (Figure 3C). However, palbociclib treatment did not prevent development of skeletal tumours; at the end of the experiment (day 35), 100% of the animals exhibited skeletal tumours in both the vehicle and the palbociclib treated group (Figure 3D). There was also no difference in the number of skeletal tumours per animal between the groups (Figure 3E), suggesting that this palbociclib schedule only inhibited the growth of established tumour colonies in bone rather than tumour cell ability to home to the bone.

### 3.4. Daily Administration of Palbociclib Is Required to Inhibit Tumour Growth in Bone

As the schedule which included a 2-day break between palbociclib cycles did not halt tumour progression in bone and in light of the short half-life of palbociclib in mice (1.5–2 h compared to 27 h in humans [39]), we next investigated the effect of daily palbociclib treatment for 4 weeks, commencing 3 days after tumour cell injection (Figure 4A). Patients with bone metastases are commonly treated with the anti-resorptive agent zoledronic acid (Zol) to inhibit tumour-induced bone destruction [40]. We therefore included a third group of animals which were treated with 100 µg/kg Zol once a week, in addition to daily palbociclib, for four weeks. We also followed tumour growth for up to 4 weeks after the end of treatment, to determine the duration of any treatment effects. As shown in Figure 4B, daily administration of palbociclib resulted in a significant inhibition of skeletal bone tumour progression throughout the 28-day treatment period, compared to the vehicle-treated animals (*p* < 0.005), with no drug-related adverse effects observed (Appendix A). During the treatment period, tumour burden was significantly lower in animals that received palbociclib, either alone or in combination with Zol, compared to the vehicle group (Figure 4C). During the treatment period, palbociclib reduced tumour growth in bone to such an extent that it could not be further reduced by addition of Zol. Palbociclib-treated animals that did not exhibit detectable hind limb tumours when imaged in vivo did display micro-metastases detectable by ex vivo imaging of isolated hind limbs (Figure 4D). On day 31 of the study, we withdrew palbociclib (±Zol) treatment and continued to monitor tumour growth. We observed an increase in detectable tumours in palbociclib (±Zol) treated animals within 10 days (Figure 4C). However, compared to animals receiving palbociclib alone, there was a trend towards reduced tumour growth in animals treated with a combination of palbociclib and Zol in the 4 weeks following cessation of palbociclib treatment. These data show that in our TNBC model, continuous daily treatment with palbociclib is required for the inhibition of overt tumour growth in bone. Though not statistically significant in our data, the trend suggests that Zol may contribute to suppressing aggressive tumour growth in bone following cessation of palbociclib +Zol combination therapy.

### 3.5. A Second Cycle of Palbociclib Fails to Perturb Overt Tumour Growth in Bone

Clinically, palbociclib is administered consecutively for 21 days followed by a 7-day break in treatment to allow for haematopoietic recovery. We therefore investigated the progression of skeletal tumours following administration of two consecutive 28-day treatment periods with palbociclib, separated by a 7-day treatment break (outlined in Figure 5A). Animals were injected with 5 × 10^4^ MDA-MB-231Luc2+GFP+ cells (i.c.), and treatment with vehicle (100 μL sodium lactate) or palbociclib (100 mg/kg) commenced 3 days later. As expected, palbociclib treatment significantly inhibited skeletal tumour growth during the first 28-day treatment cycle (Figure 5B,C). However, a significant increase in mean skeletal tumours (day 31: 0.5 vs. day 38: 1.5, *p* < 0.05) and tumour burden, and consequently number of tumour-bearing mice, was observed during the 7-day break (Figure 5C). Surprisingly, tumours continued to grow during the second palbociclib treatment cycle, despite being exposed to the same dose of drug that caused effective tumour control during the first 28-day period. The lack of responsiveness to palbociclib after a break in treatment may suggest that compensatory mechanisms, independent of the CDK4/6 pathway, are activated, allowing cells to continue progression through the cell cycle and resulting in aggressive proliferation.

### 3.6. A CDK7 Inhibitor Fails to Inhibit Growth of Palbociclib-Insensitive Tumours

We next investigated whether the addition of a different CDK inhibitor could reduce the growth of tumours that appeared to have become insensitive to palbociclib following a treatment break. CDK7 is a transcriptional cyclin dependent kinase that has been implicated in a number of breast cancer subtypes and CDK7 overexpression is associated with poor prognosis in TNBC [41,42]. As well as phosphorylating CDKs involved in the cell cycle, CDK7 is associated with core human transcription factor II (TFIIH) basal transcription complex and activates RNA polymerase II (RNA Pol II) [43]. In vitro, CDK7i induces cell cycle arrest in the MDA-MB-231Luc2+GFP+ cell line (Appendix A). When cells are exposed to palbociclib, followed by further exposure to CDK7i, a potent inhibitory effect is observed compared to vehicle (Appendix A, *p* < 0.0005). Moreover, a greater inhibitory effect is seen when palbociclib and CDK7i are used in combination following an initial exposure to palbociclib. We therefore administered the CDK7 inhibitor ICEC0942 (CDK7i) in combination with palbociclib during the second cycle of treatment to establish if inhibition of an additional CDK could inhibit growth of tumours that were no longer responding to palbociclib. We repeated the above study with the inclusion of a third group that received the CDK7i (100 mg/kg, daily), in combination with palbociclib, during the second cycle of treatment (Figure 6A). As before, we observed a significant inhibition in tumour progression in palbociclib treated animals within the first 4-week treatment cycle (Figure 6B). During the 7-day treatment break, skeletal tumours became detectable in previously tumour-free animals, indicative of resumption of tumour growth. During the second treatment cycle, palbociclib, alone or in combination with CDK7i, was unable to inhibit further overt tumour growth (Figure 6C). These data suggest that the underlying mechanisms for the continued cell cycle progression in TNBC cells lie outside the CDK7 axis and further demonstrate that the observed tumour insensitivity to treatment is not specific to palbociclib.

### 3.7. Phosphoprotein Analysis of Palbociclib-Sensitive and -Insensitive Metastatic Tumours in Bone

The MAPK pathway plays an essential role in tumorigenesis and tumour cell proliferation. Furthermore, it acts upstream, and directly activates the CDK4/6-Cyclin D axis (Figure 7A). We therefore sought to identify the potential mechanisms underlying palbociclib-insensitivity by analysing tumour cells isolated from bones of the different treatment groups using a human MAPK phosphoprotein array. This assay allowed us to explore activated (or deactivated) phosphoproteins in the MAPK signalling pathway in response to repeated palbociclib treatment. A total of 5 × 10^4^ MDA-MB-231Luc2+GFP+ cells were injected (i.c.) into 6-week-old female BALB/c nude mice, followed 2 weeks later by daily treatment with vehicle or palbociclib (100 mg/kg) (Figure 7B). This delay in treatment initiation allowed tumours to establish and grow to a measurable size in the bone (Appendix A), thus generating sufficient tumour material in bone for isolation and downstream analyses. As expected, palbociclib had a strong inhibitory effect on overt tumour growth in bone, as seen by the significant reduction in tumour burden compared to vehicle treated animals throughout the first 3-week treatment period (week 4: *p* = 0.023, week 5: *p* = 0.015, Figure 7C). Again, following a one-week treatment break, we found that growth of previously responsive tumours was not inhibited by a second cycle of palbociclib, as seen by the continued tumour growth in these mice (Figure 7C).

We then sought to investigate differences in the MAPK signalling pathway between the vehicle, palbociclib “one-cycle” and palbociclib “two-cycle” groups, by analysing tumours isolated from bones from the different treatment groups. After the end of each treatment timepoint, animals were culled and tumour-bearing hind limbs harvested. Bone marrow (BM, including tumour cells) was centrifuged and the remaining bone tissue was crushed into suspension. GFP-positive tumour cells were isolated using FACS (Figure 7D) and downstream protein analysis was conducted using a MAPK phosphorylation antibody array. We found that phosphorylation of GSK3a was significantly higher in palbociclib treated tumours (one-cycle) compared to vehicle (*p* = 0.02), demonstrating an inhibitory effect on the cell cycle during the first cycle of treatment (Figure 8). However, significant reductions in phospho-GSK3b, CREB, MEK and RSK1 were observed in tumours that were growing in the presence of palbociclib (two-cycles) compared to vehicle (*p* = 0.03, *p* = 0.04, *p* = 0.04, *p* = 0.02, respectively), suggesting that there is activation of alternative pathways compared to untreated control tumours (Figure 8). Furthermore, palbociclib-insensitive tumours (two-cycles) showed reduced phosphorylation in AKT (*p* = 0.03) and RSK2 (*p* = 0.003), as well as the tumour suppressors p38 (*p* = 0.04) and GSK3a (*p* = 0.04) and GSK3b (*p* = 0.004) compared to palbociclib-sensitive tumours (one-cycle). These results show that individual proteins in the MAPK pathway are differentially affected in TNBC tumours in bone that are sensitive to inhibition by palbociclib, compared to those that continue to grow in the presence of the drug. In addition, our data support that when CDK4/6 inhibition is discontinued, palbociclib-insensitive tumour growth is initiated.

## 4. Discussion

Due to the lack of therapeutic options and poor prognoses for TNBC patients, the search for alternative targetable pathways continues, in particular for treatment of advanced disease. CDK4/6 inhibitors improved survival in large trial and are now standard of care in metastatic ER+ve BC [44] and are also now being used in the adjuvant setting. Here, we used in vivo models to determine if CDK4/6 inhibition impacts on development and/or progression of bone metastasis, with particular emphasis on TNBC. Both TNBC and ER+ve breast cancer cell lines are sensitive to palbociclib inhibition in vitro, resulting in cell cycle arrest through inhibition of Rb phosphorylation.

Since bone is the most common site of distal breast cancer relapse [45], we initially established the effects of palbociclib on skeletal spread in ER+ve BC. In our model, palbociclib significantly inhibited ER+ve primary tumour growth in the mammary fat pad, as well as the development of bone metastases. Clinical studies have shown that palbociclib combined with endocrine therapy provides the best outcomes for patients [17], however little information exists with regard to the effects of palbociclib to impact the development and/or progression of bone metastasis. Here, we provide data to support the effectiveness of palbociclib mono-therapy to suppress development of ER+ BC metastasis, though further protection from skeletal tumour growth may be achieved in combination with hormone therapy.

We next investigated the impact of palbociclib on skeletal tumour growth in a TNBC metastasis model. Previous studies have shown that the administration (intravenously or via the intracardiac route) of a bone-homing MDA-MB-231 cell line successfully generates skeletal tumours in the hind limbs and spines of immunocompromised mice [30,46]. Utilising this model, we found that a 5 day-on/2 day-off palbociclib schedule for four weeks demonstrated potent inhibition on tumour growth in bone, resulting in a significantly lower tumour burden compared to the control group. However, all the animals in the palbociclib treated group developed skeletal tumours by the end of the experiment, suggesting that this schedule only delayed, rather than prevented, tumour progression in bone. The 5 day-on/2 day-off regimen may improve adverse events, as has been assessed in a phase II trial (NCT03007979), demonstrating reduced neutropoenia grades [47]; however no data exist on survival and disease progression using this treatment regimen. Other trials have demonstrated significant improvements in overall survival (OS) and progression free survival (PFS), with few adverse events, in response to 3 week-on/1 week-off palbociclib treatment cycles [44]. Moreover, the palbociclib half-life is 27 h in humans, in contrast to only 1.5–2 h in mice [48,49]. This, coupled with our results and data from clinical trials, suggests that the 2-day treatment break in our studies was sufficient for the tumours to escape the inhibitory effects of palbociclib, allowing tumour cells to re-enter the cell cycle and resulting in tumour progression.

Although a 3 week-on/1 week-off treatment schedule is the standard in the clinical setting, we aimed to determine the effects of continuous daily treatments with palbociclib for 4 weeks. Not only did this approach allow us to monitor possible adverse effects in response to 100 mg/kg palbociclib treatment (Appendix A), it also gave us the opportunity to observe the development of skeletal tumour growth in our metastatic model. Thus, we observed significant inhibition of skeletal tumour growth in our TNBC bone metastasis model. Once daily treatment ceased, increased skeletal tumour growth was observed in palbociclib-treated animals. Interestingly, animals treated with palbociclib combined with the anti-resorptive agent Zol displayed a lower tumour burden (though not statistically significant) during the 4 weeks after treatment cessation. Zol has been shown to inhibit bone metastasis, reduce cancer-induced bone disease and improve survival rates in preclinical and clinical studies, including in TNBC models [50,51,52]. We demonstrated that the effective inhibition of tumour growth in bone by palbociclib was unaffected by addition of Zol. Although we did not show beneficial effects on survival with palbociclib plus Zol, CDK4/6 inhibitor and Zol combination therapy may still prove to be a suitable approach in treatment of metastatic TNBC, in order to prevent development of cancer-induced bone disease and associated skeletal-related events.

In clinical use, a 1-week break in palbociclib treatment is given to allow for haematopoietic recovery. When we introduced a break from palbociclib in our studies, skeletal tumour growth rapidly resumed and was not inhibited by administration of a second cycle of palbociclib, potentially due to the outgrowth of resistant clones. Palbociclib resistance has been described in clinical studies [53,54] with a number of mechanisms that are suggested to underlie de novo resistance [55]. However, in our investigations, the lack of response to palbociclib does not fit with development of adaptive resistance, as tumour growth was always strongly inhibited when palbociclib was given daily for up to 4 weeks. Instead, we see the emergence of tumour growth that is insensitive to palbociclib once a short break (2 days) in treatment is introduced, suggesting that palbociclib-resistant clones had now emerged. We therefore attempted to target another cyclin-dependent kinase, CDK7, which is also involved in regulating the cell cycle [43,56]. CDK7 has been identified as a suitable target in drug-resistant BCs and effectively inhibits proliferation of TNBC cells in vitro [41,57,58]. Moreover, its specific targeting of tumour cells in vivo with no adverse effects on liver and kidney function were associated with the CDK7i [31]. We therefore hypothesised that inhibiting CDK7, in combination with CDK4/6 inhibition with palbociclib, could target alternative pathways and prevent palbociclib-insensitive tumour growth. However, continued tumour growth was evident when we combined palbociclib with the novel CDK7 inhibitor ICEC0942 (CDK7i) during the second cycle of treatment. In these tumours, cell cycle progression occurs in the presence of both CDK inhibitors, suggesting that the CDK7i is acting via the same pathway as palbociclib where CDK2/4/6 inhibition are no longer relevant in palbociclib-insensitive cells.

We next explored the potential mechanisms involved in palbociclib-insensitivity, focussing on increased activation of the MAPK pathway, which has been previously implicated in palbociclib resistance [59]. Proteins and signalling molecules involved in the MAPK pathway act upstream of and directly activate the CDK4/6-Cyclin D axis [60]. We therefore sought to assess the MAPK signalling pathway in tumours that were sensitive (preceding the treatment break) and insensitive (following the treatment break) to palbociclib. We did not observe any significant differences in phosphorylated-MAPK proteins between vehicle-treated tumours and palbociclib-sensitive tumours. This is expected as the MAPK pathway is not activated in palbociclib-induced cell cycle arrest. Although we observed a significant increase in ERK1/2 in palbociclib-sensitive tumours (compared to vehicle), this could be a compensatory mechanism to overcome cell cycle arrest; however, since the CDK4/6 pathway is downstream of this, palbociclib still exerts its inhibitory effect. Our most significant finding was a reduction in the phosphorylation of p38 in palbociclib-insensitive tumours compared to those that were palbociclib-sensitive. p38 is considered to be a tumour suppressor; its phosphorylation leads to cell cycle arrest [61]. Our data therefore suggest that the reduced phosphorylation of p38 prevents p38-driven inhibition of on the cell cycle and thus may result in growth of palbociclib-insensitive tumours. However, the MAPK pathway is a complex, interconnected signalling cascade, dependent on extracellular signals as well as crosstalk from parallel pathways. For example, RSK2 intersects the MAPK- and PI3K pathways [62]. RSK2 was discovered to have a critical role in TNBC viability whereby silencing RSK2 led to TNBC growth inhibition and apoptosis [63,64]. Phosphorylated RSK2 activates cytoplasmic and nuclear molecules that regulate tumorigenesis and cell division [65,66,67]. In the current study, we observed a significant reduction in phosphorylated RSK2 in palbociclib-insensitive tumours, contrary to the known functional dependency of RSK2 in TNBC. We also observed reduced phosphorylated AKT in these tumours, another important cell cycle regulator within the PI3K pathway where activation is implicated in drug resistance [68]. The reduced phosphorylation of these proteins suggest that palbociclib-insensitive tumour cells continue to grow through the lack of p38 control mediated by pathways outside the MAPK and PI3K axes. This, along with our CDK7i data, suggests that palbociclib-insensitive cells continue to grow through alternative pathways that allow activation of the cell cycle.

Our data, in combination with those previously published, highlight the complexity of signalling cascades, not only in tumorigenesis but also in drug-insensitive tumour growth, whereby alternative pathways may be activated to maintain tumour cell proliferation and survival. Extracellular signals such as growth factors, cytokines and stress stimulate tyrosine kinase receptors (TRK) and/or G coupled protein receptor (GPCRs), leading to the activation of the MAPK and PI3K pathways [69]. Our tumour cells were harvested from the hind limbs of tumour-bearing mice; it is therefore important to consider the vast array of signalling molecules that are interacting with tumour cells, and vice versa. For example, tumour cells establish in areas with high turnover where there is an increased number of active osteoclasts [70], and highly complex interactions between tumour cells and the surrounding microenvironment support tumour cell survival and growth [71]. Interestingly, it has been shown that MDA-MB-231-derived factors activate calcium/protein kinase C and TGF-β dependent ERK1/2 and p38 pathways in osteoclasts [72]. Furthermore, bone microenvironment-derived TGF-β is involved in maintaining tumour cell survival and growth, demonstrating a complex bidirectional function of TGF- β [73]. TGF-β (and other bone derived factors such as BMP) activates the SMAD pathway, another vital pathway in tumorigenesis in the bone [74,75]. This, along with the lack of effect seen by addition of CDK7 inhibition, emphasises the complexity of signalling pathways involved in palbociclib-insensitivity in models of bone metastasis. Therefore, further analysis of palbociclib-insensitive cells would provide an insight in determining whether resistance is tumour-intrinsic or, as described above, a function of the microenvironment. Since many of these pathways act upstream of the CDK4/6-Rb axis, our data also support the exploration of alternative pathway targets that can be combined with CDK4/6 inhibitors.

## 5. Conclusions

In summary, we demonstrated a novel effect of the CDK4/6 inhibitor, palbociclib, in inhibiting the progression of BC metastasis in the bone as a monotherapy. We further showed that continuous palbociclib treatment is essential in maintaining its inhibitory effect on metastatic TNBC growth in the bone. The introduction of a one-week break in treatment allows tumour cells to escape the effects of palbociclib, thus making them insensitive to a second treatment. Proteins involved in the MAPK pathways may be implanted in palbociclib-insensitivity, giving rise to new therapeutic opportunities for the treatment of CDK4/6-insensitive BC.

## Figures and Tables

**Figure 1 cancers-15-02211-f001:**
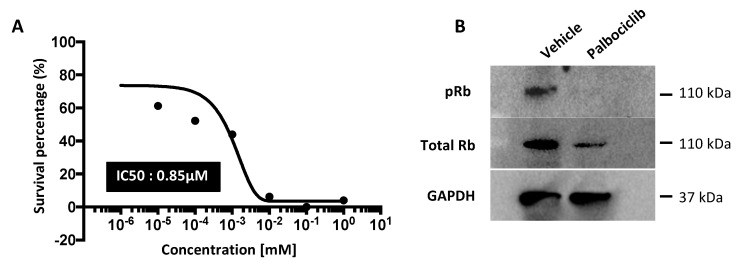
Palbociclib induces cell cycle arrest by inhibiting the phosphorylation of Rb in TNBC cell line. (**A**) Triple negative MDA-MB-231Luc2+GFP+ human breast cancer cells were treated with increasing concentrations (1 nM–1 mM) of palbociclib for 72 h after which an MTT proliferation assay was conducted. (**B**) Cells were treated with palbociclib at IC_50_ (0.85 μM) or vehicle (sodium lactate) for 72 h and harvested for downstream analyses. Cell lysates were prepared for Western blot for the detection of pRb and total Rb protein. The original Western blot can be found at Appendix A. (**C**) Flow cytometry analysis of pRb protein (mean ± SEM compared to vehicle (*n* = 3, *** *p* < 0.005; *t*-test). (**D**,**E**) Cells were treated with 0.85 μM dose palbociclib or vehicle for 72 h and cell cycle analysis was conducted at the indicated time points after drug removal (*n* = 3, **** *p* < 0.0005; *t*-test).

**Figure 2 cancers-15-02211-f002:**
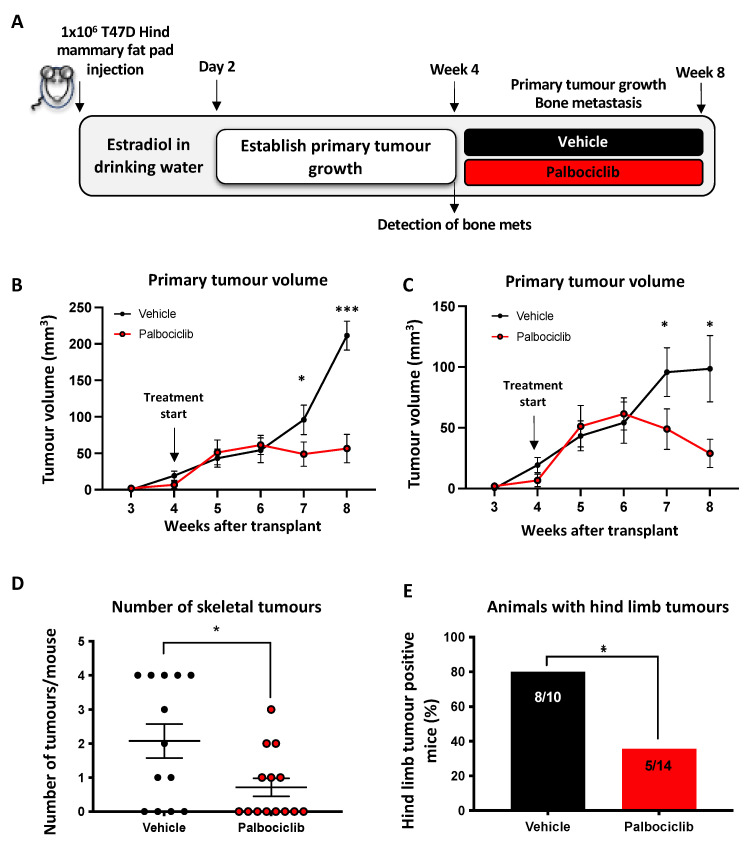
Palbociclib treatment inhibits primary tumour growth and bone metastasis in ER+ BC. (**A**) Generation of ER+ BC model of metastasis. The group of 14-week-old BALB/c nude mice were supplemented with estradiol in the drinking water. Two days later, 1 × 10^6^ T47D(luc2+GFP+) ER+ cells were injected bilaterally into the hind mammary fat pad. An amount of 100 mg/kg palbociclib was administered 4 weeks after tumour cell injection and continued for a further 4 weeks. (**B**) Primary tumour volume in vehicle treated animals compared to palbociclib from our initial experiment (*n* = 7 per group, mean ± SEM, * *p* < 0.05, *** *p* < 0.0005). (**C**) Primary tumour volumes from two experiments combined (vehicle (*n* = 14) vs. palbociclib (*n* = 14) mean ± SEM, * *p* < 0.05). (**D**) The number of skeletal tumours after four weeks of treatment in the two studies combined (mean ± SEM *n* = 13 per group * *p* < 0.05). (**E**) The proportion of mice exhibiting skeletal tumours in response to palbociclib after four weeks of treatment in the two studies (mean ± SEM, * *p* < 0.05).

**Figure 3 cancers-15-02211-f003:**
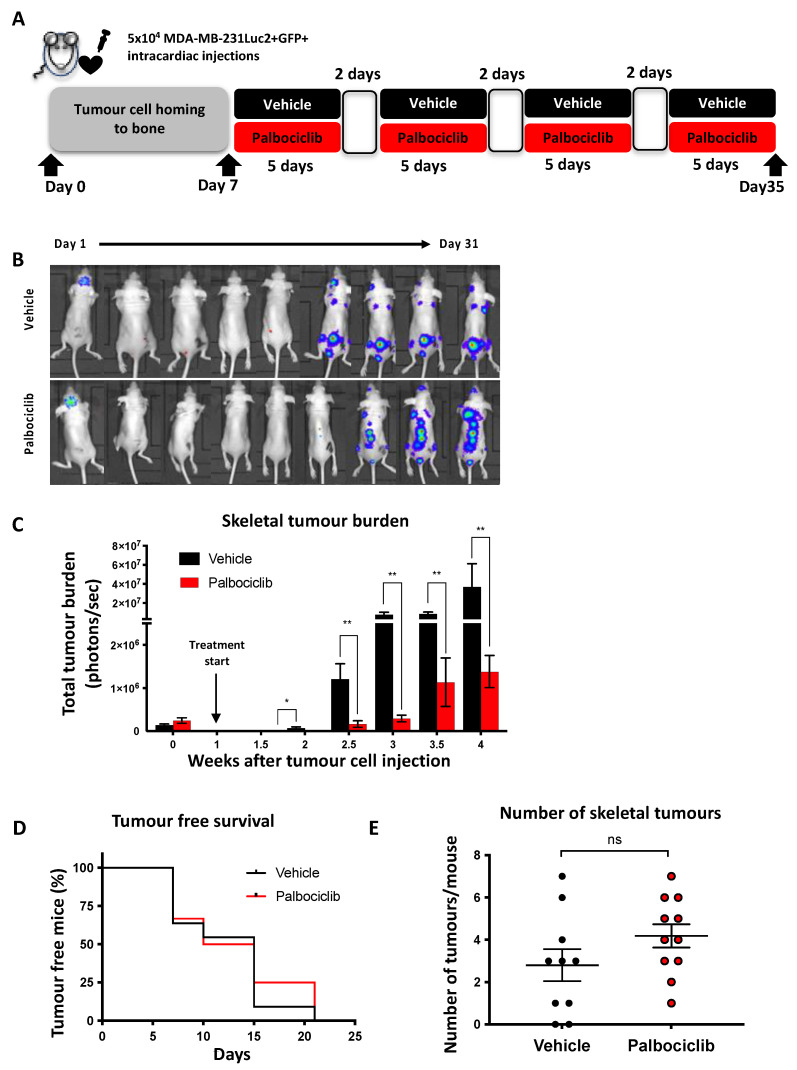
A 5-day-on/2-day-off palbociclib treatment reduces metastatic tumour burden in bone. (**A**) Schematic outline of the experiment investigating effects of a 5-day-on/2-day-off palbociclib treatment. Mice were injected with 5 × 10^4^ MDA-MB-231Luc2+GFP+ cells via the intracardiac route. Treatment with 100 mg/kg palbociclib or vehicle commenced after 7 days and continued 5 days/week for 4 cycles. (**B**) Example of in vivo images demonstrating progression of skeletal tumours in both control and palbociclib treated animals. (**C**) Progression of skeletal tumour growth measured as tumour burden (photons/s) using in vivo live imaging (Mean ± SEM *n* = 10 (control) vs. *n* = 11 (palbociclib), * *p <* 0.05 ** *p <* 0.005). (**D**) Tumour-free mice throughout the duration of the study. (**E**) Number of tumours per mouse at day 35 (Mean ± SEM *n* = 10 (control) vs. *n* = 11 (palbociclib)). ns: not significant.

**Figure 4 cancers-15-02211-f004:**
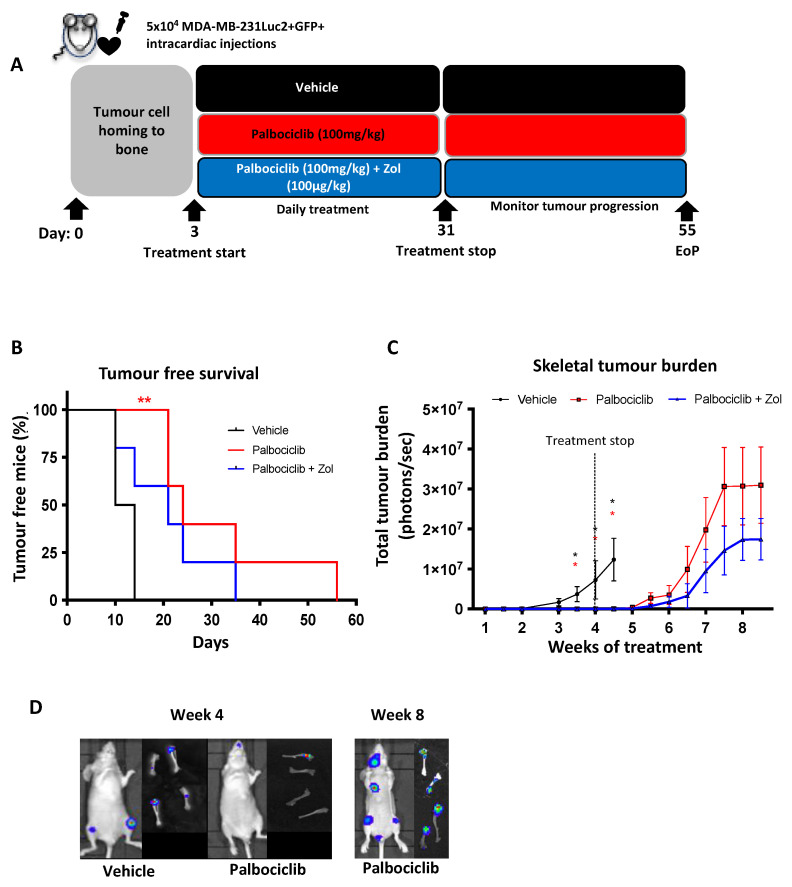
Continuous Palbociclib treatment inhibits tumour growth in bone which is not improved by zoledronic acid. (**A**) Schematic outline of the experiment. Mice were injected with 5 × 10^4^ MDA-MB-231Luc2+GFP+ cells via the intracardiac route. Treatment with 100 mg/kg palbociclib or vehicle commenced on day 3 and continued daily for 28 consecutive days. A third group received palbociclib daily in addition to the anti-resorptive agent zoledronic acid (Zol) once a week (intraperitoneally). (**B**) Detection of skeletal tumours in vehicle and palbociclib ± Zol treated animals throughout the duration of the study (*n =* 8 per group, (red) ** *p <* 0.005; Log-rank (Mantel–Cox) test vehicle vs. palbociclib). (**C**) Tumour progression as measured by tumour burden in response to vehicle or palbociclib ± Zol (Mean ± SEM *n* = 8 per group * *p <* 0.05). (**D**) Representative images of tumour progression at week 4 and week 8 in the live mouse and detected by ex vivo imaging of isolated hind limbs. EoP—End of procedure.

**Figure 5 cancers-15-02211-f005:**
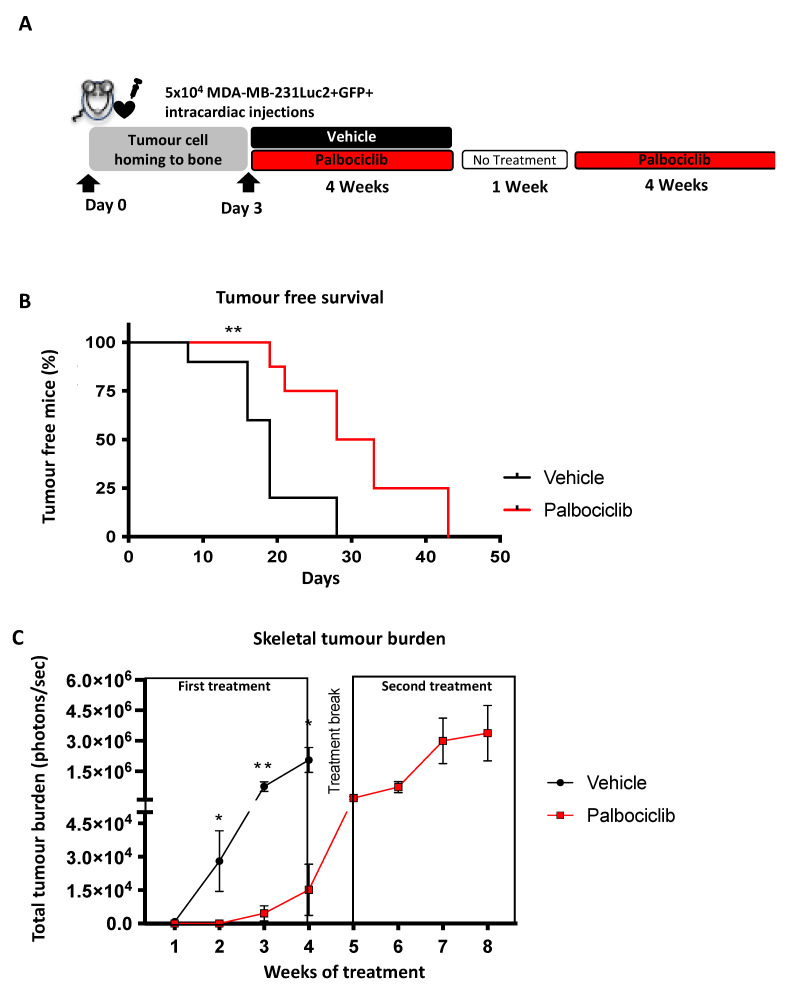
Palbociclib fails to inhibit tumour growth in bone following a break in treatment. (**A**) Schematic outline of the experiment. Mice were injected with 5 × 10^4^ MDA-MB-231Luc2+GFP+ cells via the intracardiac route. Treatment with 100 mg/kg palbociclib or vehicle commenced on day 3 and continued for 28 consecutive days, followed by a 7-day break before a second cycle of daily palbociclib treatment commenced. (**B**) Percentage of tumour positive mice (*n =* 8 per group, ** *p* < 0.005; Log-rank (Mantel–Cox) test vehicle vs. palbociclib) and (**C**) mean (±SEM) tumour burden in each treatment arm (*n* = 8 per group, * *p* < 0.05, ** *p* < 0.005; multiple *t*-test).

**Figure 6 cancers-15-02211-f006:**
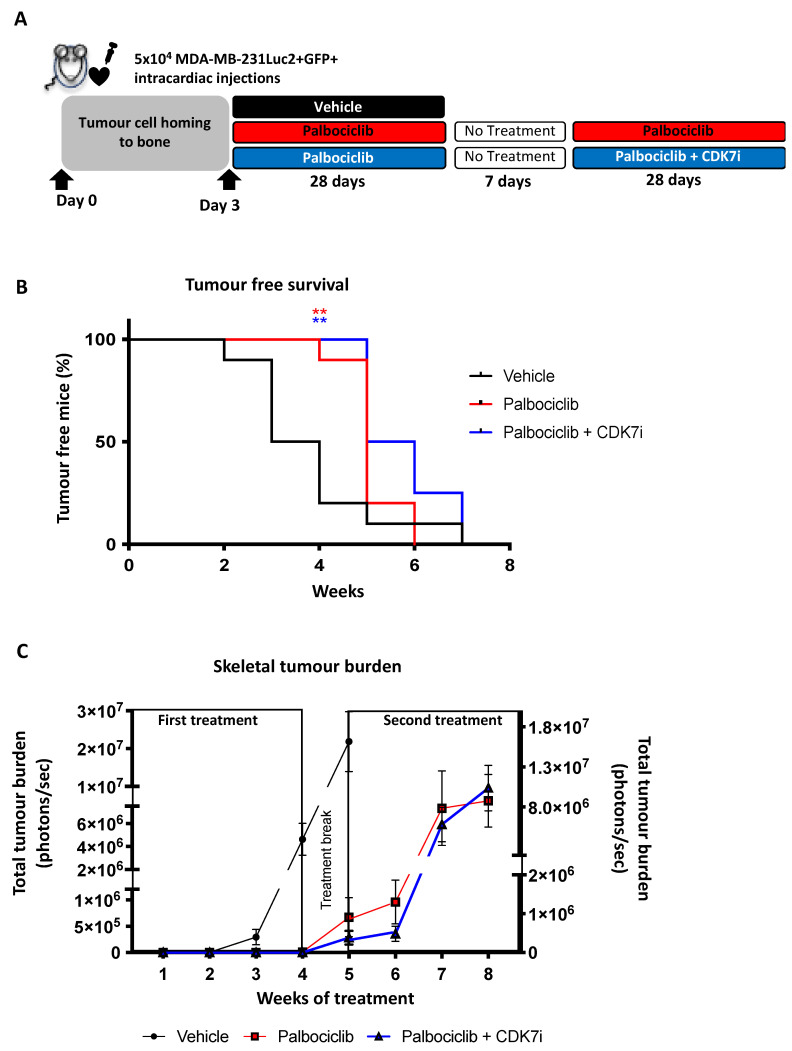
CDK7i in combination with palbociclib is insufficient in inhibiting palbociclib treated skeletal tumours. (**A**) Schematic outline of the experiment. Mice were injected with 5 × 10^4^ MDA-MB-231Luc2+GFP+ cells via the intracardiac route. Treatment with 100 mg/kg palbociclib or vehicle commenced on day 3 and continued daily for 28 consecutive days followed by a 7-day break and a second cycle of treatment consisting of palbociclib alone or in combination with 100 mg/kg of the CDK7 inhibitor (ICEC0942) for a further 28 days. (**B**) Percentage of tumour positive mice (*n =* 8 per group, ** *p* < 0.005; Log-rank (Mantel–Cox) test vehicle vs. palbociclib ± CDK7i. (**C**) tumour burden in each treatment arm (mean ± SEM) (*n* = 8 per group).

**Figure 7 cancers-15-02211-f007:**
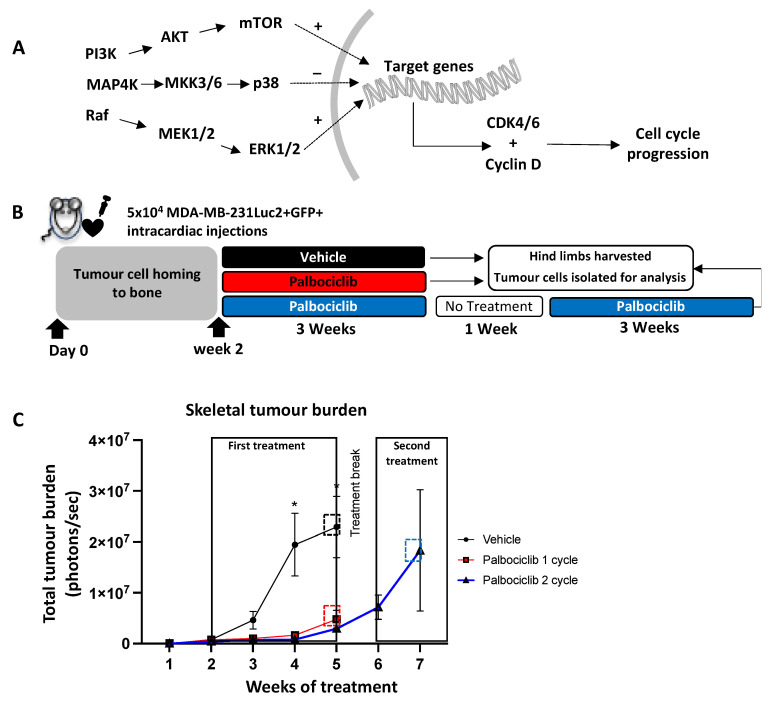
Palbociclib inhibits established metastatic tumour growth in bone, but growth continues after a treatment break. (**A**) Proteins involved in the MAPK pathway activate their downstream targets by phosphorylation. This leads to the translocation of specific phosphoproteins that activate target genes which in turn promote the production of cyclin D and CDK4/6 resulting in cell cycle progression and survival. (**B**) Schematic outline of the experiment. Mice were injected with 5 × 10^4^ MDA-MB-231Luc2+GFP+ cells via the intracardiac route. Treatment with 100 mg/kg palbociclib or vehicle commenced on day 14 and continued for 21 consecutive days, followed by a 7-day break before a second cycle of daily palbociclib treatment commenced. (**C**) Tumour burden in each treatment arm (*n* = 8 per group, * *p* < 0.05; multiple *t*-test) (**D**) FACS gating strategy for the isolation of GFP+ tumour cells from whole bone and bone marrow, and downstream phosphoprotein array.

**Figure 8 cancers-15-02211-f008:**
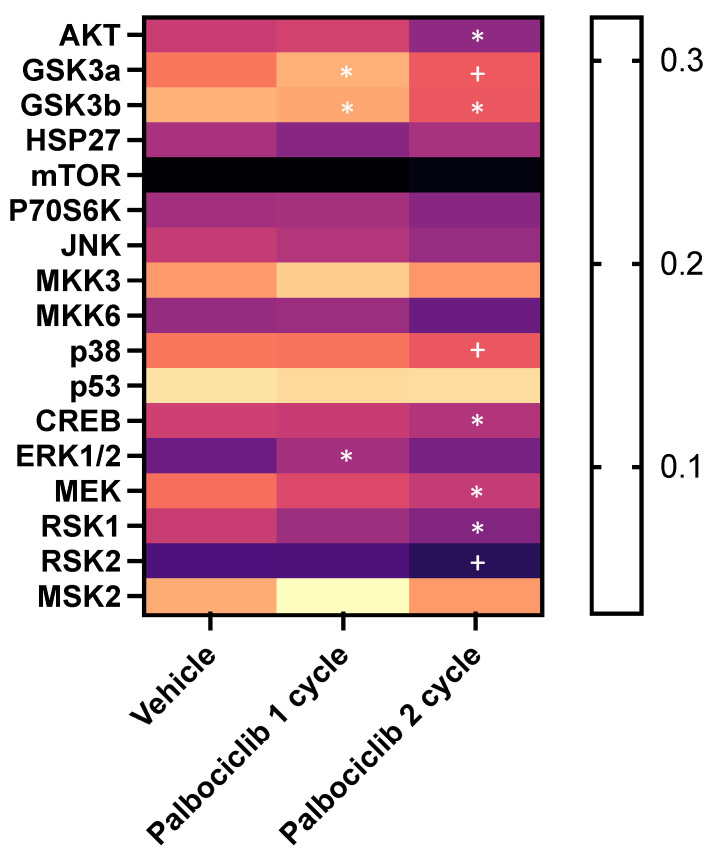
The effect of one- and two-cycles of palbociclib treatment on the MAPK pathway. Heatmap showing the levels of phosphoproteins from tumour cells isolated from vehicle-, one-cycle palbociclib- and two-cycle palbociclib-treated animals. Expression levels measured as band intensities (* *p* < 0.05 compared to vehicle, + *p* < 0.05 compared to palbociclib 1 cycle).

## Data Availability

The data presented in this study are available on request from the corresponding author.

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
