# Peer review of "The CDK4/6 Inhibitor Palbociclib Inhibits Estrogen-Positive and Triple Negative Breast Cancer Bone Metastasis In Vivo"

_cancers, 2023, doi:10.3390/cancers15082211_

Round 1
Reviewer 1 Report
In this work, Saleh et al. focus on the specificity of CDK4/6i to treat bone metastases from ER+ BCa and TNBC. CDK4/6i are approved to treat metastatic ER+ BCa, and the vast majority of these patients will present with bone metastases. Therefore, CDK4/6i efficacy in this context is known and data does not bring novelty. On contrary, in TNBC, there are few studies and this part of the work is more interesting. When reading the title, abstract and objectives, it sounds a little confusing what was really the purpose, and why ER+ BCa was added in particular.
There was only one in vivo model of each tumor type, T47D and MDA-231 for ER+ and TNBC, respectively. These two cell lines, in particular, are very sensitive to Palbociclib according to published data; and data presented in Figure 1 is according to other works. Supp Figs were not available.
The characterization of treated tumors was done only by MAPK phosphoprotein analysis, which highlighted p38 as mediator of resistance.
In ER+ BCa CDK4/6i require combination with endocrine therapy, however palbociclib was used as monotherapy.
Estradiol was added to the drinking water, instead of using s.c pellets, which although decreasing toxicity can lead to uneven tumor growth of ER+ BCa cells. In Fig 2B and 2C is clear that not all mice developed primary tumors as there are tumors with 0mm3 in the VEH group, therefore it should be explained why they were kept in the experiment after randomization at week 4. Actually, it should be referred how randomization was conducted and the tumor vol at that point.
For primary tumors a growth curve could be presented, clarifying for example if tumors regressed (there several 0mm3 in palbo group also), or if therapy delayed growth. In this model was Palbo given daily for 4 weeks or 5 days ON / 3 OFF?
As authors mention, clinically palbociclib is administered consecutively for 21 days followed by a 7-day break. Therefore, why was not this the scheme chosen for in vivo experiments? Only in Fig 7B this was the therapeutic scheme.
The authors addressed the use of BP (zoledronic acid) to treat tumors with acquired resistance after one therapy cycle, without showing benefit. How was BP effect on bone remodeling assessed?
Authors chose a CDK7 inhibitor, although CDK7 has been associated with resistance to CDK4/6i, there are many other reported drivers of resistance. It would be important to measure CDK7 expression in Palbo-treated cells and tumors; and to show that CDK7i is inhibiting CDK7.
The dosage of Palbo was very high (100mg/Kg) and for some experiments the treatment was quite long. It would be important to show mice body weight on treatment for all experimentys, to assess toxicity.
Other aspects:
There is no Ethical Statement regarding the approval of the in vivo experiments.
Lines 212-214 to be removed.
Intensity Legend of Fig 8 is missing.
Author Response
- In this work, Saleh et al. focus on the specificity of CDK4/6i to treat bone metastases from ER+ BCa and TNBC. CDK4/6i are approved to treat metastatic ER+ BCa, and the vast majority of these patients will present with bone metastases. Therefore, CDK4/6i efficacy in this context is known and data does not bring novelty. On contrary, in TNBC, there are few studies and this part of the work is more interesting. When reading the title, abstract and objectives, it sounds a little confusing what was really the purpose, and why ER+ BCa was added in particular.
Response: We are pleased that the reviewer recognised the novelty of our study with regards to TNBC. The reason for including an ER+ve model was to demonstrate the effect of palbociclib on tumour growth in bone, irrespective of primary tumour type. The ER+ve model serves as a ‘positive control’ to show utility in ER+ve bone metastasis and to strengthen our study which would otherwise have been limited to a single model, something that is generally not accepted by quality journals. We have now altered the final statement of the introduction to better direct the reader (line 84-88)
- There was only one in vivo model of each tumor type, T47D and MDA-231 for ER+ and TNBC, respectively. These two cell lines, in particular, are very sensitive to Palbociclib according to published data; and data presented in Figure 1 is according to other works. Supp Figs were not available.
Response: Figure 1 is proof of concept that palbociclib inhibits the cell line that we intended to use in vivo. We are unsure why the supplementary figures appeared to be unavailable to the reviewer as these were submitted using the online portal together with the other manuscript files.
- The characterization of treated tumors was done only by MAPK phosphoprotein analysis, which highlighted p38 as mediator of resistance.
Response: As stated in line 535-537, we aimed to identify molecules that may have an import role in palbociclib insensitivity. This approach was a ‘screening’ method which could lead to further investigations in insensitivity and possibly resistance.
- In ER+ BCa CDK4/6i require combination with endocrine therapy, however palbociclib was used as monotherapy.
Response: The reviewer is correct in that CDK4/6 inhibitors are combined with endocrine therapy in breast cancer, a point we also make in our manuscript. In this study, we sought to demonstrate the efficacy of palbociclib alone in models of breast cancer growth in the bone to indicate efficacy of CDK4/6 inhibition against bone metastasis. Addition of endocrine therapy to the ER+ve model would have reduced tumour growth and prevented the progression of bone metastases, making it difficult to establish the effect of palbociclib alone. As the main focus of our work was on TNBC, the ER+ model served as a positive control only.
- Estradiol was added to the drinking water, instead of using s.c pellets, which although decreasing toxicity can lead to uneven tumor growth of ER+ BCa cells. In Fig 2B and 2C is clear that not all mice developed primary tumors as there are tumors with 0mm3 in the VEH group, therefore it should be explained why they were kept in the experiment after randomization at week 4. Actually, it should be referred how randomization was conducted and the tumor vol at that point.
Response: We show that mice establish metastatic tumours in the bone at around 4/5 weeks (as shown in supp fig 2 that the reviewer was unable to see), naturally some mice will develop tumours later in the studies, thus no animals were removed from the study. Furthermore, stratified randomisation was conducted as queried by the reviewer, this detail has now been included in the manuscript (line 108).
- For primary tumors a growth curve could be presented, clarifying for example if tumors regressed (there several 0mm3 in palbo group also), or if therapy delayed growth. In this model was Palbo given daily for 4 weeks or 5 days ON / 3 OFF?
Response: As the reviewer suggests, we have revised figure 2 to show the effect of palbociclib on primary tumour growth from 2 independent studies (panel B: the initial study, panel C: our 2 studies combined). These data show that primary tumour burden in the palbociclib-treated group exhibit a small reduction in primary tumour size, consistent with a cytostatic effect. In contrast, tumour burden in the vehicle-treated group increases throughout the experimental period. We have also described these findings in lines 334-338 As specified in line 338, palbociclib was administered daily for 4 weeks in this experiment.
- As authors mention, clinically palbociclib is administered consecutively for 21 days followed by a 7-day break. Therefore, why was not this the scheme chosen for in vivo experiments? Only in Fig 7B this was the therapeutic scheme.
Response: As our study was the first to investigate the effect of palbociclib in these models, a cautious approach to dosing frequency (5 days on-2 days off) was initially taken to ensure that the drug was tolerated over the 4-week experimental period. Once we had established tolerability, we removed the 2-day break in treatment in order to determine if continuous treatment was effective. Furthermore, we commenced treatment 2-weeks after intracardiac injection of the tumour cells, this significantly extended the tumour-bearing time of the mice. We therefore aimed to shorten the study while, at the same time, observing the inhibitory effects of palbociclib on metastatic tumour growth in the bone.
- The authors addressed the use of BP (zoledronic acid) to treat tumors with acquired resistance after one therapy cycle, without showing benefit. How was BP effect on bone remodeling assessed?
Response: We did not assess bone remodelling in this study, as this has been comprehensively demonstrated in our previous work using a single 100ug/kg dose of Zol (Haider MT et al Bone. 2014 Sep;66(100):240-50. doi: 10.1016/j.bone.2014.06.023, Brown HK et al J Bone Oncol. 2012 Jun 19;1(2):47-56. doi: 10.1016/j.jbo.2012.05.001.) The impact of palbociclib on metastatic tumour growth in bone was the sole aim of our study, not the effect on cancer-induced bone disease.
- Authors chose a CDK7 inhibitor, although CDK7 has been associated with resistance to CDK4/6i, there are many other reported drivers of resistance. It would be important to measure CDK7 expression in Palbo-treated cells and tumors; and to show that CDK7i is inhibiting CDK7.
Response: We have shown that palbociclib and the CDK7i elicit potent inhibitory effects on cell proliferation in vitro with a greater effect observed in combination with palbociclib. These data are now added (lines 501-506) to the manuscript as a new supplementary figure 5.
- The dosage of Palbo was very high (100mg/Kg) and for some experiments the treatment was quite long. It would be important to show mice body weight on treatment for all experiments, to assess toxicity.
Response: We respectfully disagree with the reviewer that the dose used is very high, published studies have used 120-150 mg/kg/day (Cook Sangar ML et al. Clin Cancer Res. 2017 Oct 1;23(19):5802-5813. doi: 10.1158/1078-0432.CCR-16-2943, Yamamoto, Takuro et al. Breast cancer research and treatment vol. 174,3 (2019): 615-625. doi:10.1007/s10549-018-05104-9). In addition, the half-life of palbociclib in mice is 1.5-2h, meaning that exposure times to 100mg/kg are relatively short in vivo (Smith D et al J Chromatogr B Analyt Technol Biomed Life Sci. 2011 Oct 1;879(27):2860-5. doi: 10.1016/j.jchromb.2011.08.009). As stated in line 409 of the manuscript, we did not observe any adverse effects of palbociclib in any of our experiments. Importantly, the mean body weight of animals receiving 100mg/kg palbociclib daily for 4 weeks (17g) was comparable to that of the vehicle treated group (18g), demonstrating no negative impact of palbociclib.
Other aspects:
- There is no Ethical Statement regarding the approval of the in vivo experiments.
Response: This was given under the “Institutional Review Board Statement” (line 804-806)
- Lines 212-214 to be removed.
Response: These lines have been removed as requested by the reviewer.
- Intensity Legend of Fig 8 is missing.
Response: We believe this is a formatting issue that has now been resolved.
Reviewer 2 Report
In this study authors attempt to assess CDK4/6 inhibitor Palbociclib effectiveness in estroge receptor-positive and triple negative breast cancer bone metastasis by in vivo models. As cell line model, authors use MDA-MB-231 TNBC and T47D ER+ ve BC.
Overall, this study could be considered an intriguing and valid study in the attempt of increasing our knowledge in the mechanism of action and effectiveness of these compounds in the inhibition of bone metastasis from breast cancer.
However, it suffers of important experimental bias and determinant controls that leads the conclusion reached by the authors not well demonstrated. Indeed, it requires a more deeply and comprehensive investigation and statistical analysis in order to confirm the effect of the CDK4/6 inhibitor, palbociclib, in inhibiting the progression of breast cancer bone metastasis.
Major revisions should be done prior resubmission to evaluate a potential publication on Cancers.
Major comments:
The in vivo models breast cancer bone metastasis model performed by the authors are not clear. Authors should explain the reasons why for ER+ve model they used a spontaneous breast cancer metastasis model by cell injection in the mammary fat pad, instead for TNBC a metastatic outgrowth model in bone by intracardiac model. Indeed, these two different injections enable to evaluate different stages of bone metastasis (http://dx.doi.org/10.20517/2394-4722.2021.14; https://doi.org/10.1016/B978-0-12-416721-6.00043-1).
In order to evaluate in a more comprehensive analysis the effect of palbociclib, authors should perform the two different models for both cell lines in order to evaluate at which time point of the metastatic progression the drug could be more efficient. Also, for the same reason, in 312-313 line they could not say that “palbociclib…did not prevent the early stages of bone metastasis” since the model by intracardiac injection is not the ideal to evaluate early stages, but only the tumor cell growth and the latest stages of bone metastasis (interactions of tumor cells with bone microenvironment). In order to see early stages, authors should perform a spontaneous bone metastasis model.
Authors should explain the reasons why for the in vivo models they used only MDA-MB-231 (TNBC) and T47D cell lines meanwhile for the western blot analysis they add also MDA-MB-468 (TNBC) and MCF7 cell lines. In this way, the results obtained by the in vivo experiments could be cell-line specific.
Authors should explain why for TNBC model they used 6-week-old BALB/c mice, meanwhile for ER+ve BC model they used 14-week-old BALB/C mice. Indeed, in this last scenario, 14-week-old mice seem pretty old mice to perform spontaneous bone metastasis model. Moreover, they should specify in material and methods the site of drug injection.
Fig.1 Authors should always specify the cell line to which the results refer in the figure legends
Fig.1A MTT analysis does not inform about the percentage of cells, instead it inform about cell viability and citotoxicity (authors could refer to this paper: https://doi.org/10.3390/ijms18081655). Authors should change it in the graph as survival percentage (%).
Fig.1B Authors assume that Palbociclib treatment leads a decrease in the phosphorilation of Rb protein in MDA-MB-231. However, looking at the western blot analysis, there is a significant decrease of the total Rb protein. Authors should analyze the % decrease of pRb compared to the total Rb in order to demostrate the effect induced by Palbociclib. Authors should also discuss the difference in total Rb expression.
Fig. 1E Are these data not significant? Authors should explain
Fig.2A There is a discrepancy in the number of T47D cells injected compared to the number mentioned in the material and methods section (5X10^4 cells). Indeed, in the figure it is reported 1x10^6 cells. Authors should standardize this data.
The addition of estradiol could affect the primary tumor growth and bone metastasis in the model. Authors should explain whether this compound could affect the results obtained and the Palbociclib effect.
Fig. 2C Authors report the number of skeletal tumors. However this data is not clear, indeed it doesn’t seems authors performed x-rays analysis or µCT to evaluate any skeletal metastasis, which is a consequence of bone homeostasis imbalance induced by tumor cells and the establishment of osteolytic area (in the case of breast cancer). For this reason, authors cannot be sure that cells establish in this model bone metastasis and the data obtained by bioluminescent analysis can only be referred to cell growth capability in the bone. In order to demostrate that cells are able to establish bone metastasis, authors should perform at least x-rays analysis to evaluate any osteolytic area and histology and H&E staining of the bone samples could add information about tumor cells and how they affect bone microenvironment (authors could refer to this paper: doi: 10.1177/0300985815586223 and to this paper https://doi.org/10.1038/s41556-021-00641-w in order to check how to report in the figure the bone metastasis establishment in in vivo experiments). These should be done for all the in vivo experiments in order to demostrate that cells establish bone metastasis and that Palbociclib can affect this process. Otherwise, it seems that the drug can affect only cell growth and skeletal tumor burden.
Authors should explain better the rationale of the different schedule treatment of Palbociclib.
In the introduction, authors should better describe the use of bone targeted agents in bone metastatic patients. Later, they should better explain the rationale for the experiment performed to evaluate the combination of Palbociclib and Zoledronic Acid.
Fig.7 This figure needs an higher resolution because it is difficult to read
Fig.8 This figure needs statistical analysis.
Author Response
- The in vivo models breast cancer bone metastasis model performed by the authors are not clear. Authors should explain the reasons why for ER+ve model they used a spontaneous breast cancer metastasis model by cell injection in the mammary fat pad, instead for TNBC a metastatic outgrowth model in bone by intracardiac model. Indeed, these two different injections enable to evaluate different stages of bone metastasis.
Response: The reviewer correctly points out that the two models we have used to explore the effect of palbociclib on bone metastasis reflect slightly different stages of disease progression. The TNBC intracardiac injection model does not involve dissemination from a primary tumour site, but still captures the early stages of single cell dissemination to and colonisation of bone niches, in addition to subsequent metastatic progression. The ER+ve model includes spread from a mammary fat pad tumour to the skeleton. Importantly, we only initiated palbociclib treatment in the ER+ model once bone metastases were detectable (week 4 following tumour cell injection), hence in both models the therapeutic effect was on tumours development in (and not dissemination to) bone. Earlier intervention in the ER+ve model would allow palbociclib to (potentially completely) inhibit primary tumour growth and reduce dissemination, meaning that we would not have been able to distinguish effect of the agent on tumour growth in bone. This has been clarified by expanding the relevant text in the materials and methods (line 188 onwards) to better explain the rationale of using these two models.
- In order to evaluate in a more comprehensive analysis the effect of palbociclib, authors should perform the two different models for both cell lines in order to evaluate at which time point of the metastatic progression the drug could be more efficient. Also, for the same reason, in 312-313 line they could not say that “palbociclib…did not prevent the early stages of bone metastasis” since the model by intracardiac injection is not the ideal to evaluate early stages, but only the tumor cell growth and the latest stages of bone metastasis (interactions of tumor cells with bone microenvironment). In order to see early stages, authors should perform a spontaneous bone metastasis model.
Response: The reviewer raises an interesting point; however, the focus of our studies was to determine if palbociclib could inhibit progression of bone metastasis following tumour cell colonisation of bone (and not the initial stages of dissemination to bone). Our aim was to generate supporting data for the potential use of palbociclib in patients with confirmed bone metastasis, rather than in the adjuvant setting. To clarify, by ‘early stages’ of bone metastasis we mean the growth from single cells entering bone (as is the case following intra-cardiac injection), whereas the reviewer interprets early stages as dissemination from a primary tumour site. The manuscript text has been revised to emphasise this point (lines 329 and 394-395).
- Authors should explain the reasons why for the in vivo models they used only MDA-MB-231 (TNBC) and T47D cell lines meanwhile for the western blot analysis they add also MDA-MB-468 (TNBC) and MCF7 cell lines. In this way, the results obtained by the in vivo experiments could be cell-line specific.
Response: We used the above cell lines in vivo as they have been proven as working models of bone-homing metastatic disease, and we wanted to establish the effects of palbociclib in both ER+ve and TNBC models whilst adhering to the principles of the 3Rs (hence not repeat studies in additional models as we had shown effect in 2 different models already). Furthermore, we aimed to demonstrate as proof of principle that palbociclib induces cell cycle arrest in alternative cell lines, in order to show that the effects of palbociclib in our study are not cell line specific as previously published (https://doi.org/10.1186/bcr2419). We have removed the data from the MDA-MB-468 cell lines to avoid confusion.
- Authors should explain why for TNBC model they used 6-week-old BALB/c mice, meanwhile for ER+ve BC model they used 14-week-old BALB/C mice. Indeed, in this last scenario, 14-week-old mice seem pretty old mice to perform spontaneous bone metastasis model. Moreover, they should specify in material and methods the site of drug injection.
Response: We apologise if this was unclear in our manuscript. These are the critical ages for bone development and turnover to allow suitable conditions for tumour cells to home to the bone and grow, with 6-week old animals being the standard for the intracardiac injection model. This is now explained in the revised manuscript (lines 183-186). Details of drug treatments are already described under “in vivo studies” section (lines 199-206 and 212) and have been clearly emphasised (line 124). Importantly, 80% of animals in the ER+ve model developed bone metastasis, in line with our previous studies using 12-week old animals (see figure 1c, Holen et al Clin Exp Metastasis. 2016 Mar;33(3):211-24. doi: 10.1007/s10585-015-9770-x.), demonstrating that the age of the mice (14 weeks) did not preclude development of bone metastases.
- 1 Authors should always specify the cell line to which the results refer in the figure legends
Response: We are unsure as to what the reviewer means as this was the case in our manuscript.
- 1A MTT analysis does not inform about the percentage of cells, instead it inform about cell viability and cytotoxicity (authors could refer to this paper: https://doi.org/10.3390/ijms18081655). Authors should change it in the graph as survival percentage (%).
Response: We thank the reviewer for pointing this out and have changed the figure and its description accordingly.
- 1B Authors assume that Palbociclib treatment leads a decrease in the phosphorilation of Rb protein in MDA-MB-231. However, looking at the western blot analysis, there is a significant decrease of the total Rb protein. Authors should analyze the % decrease of pRb compared to the total Rb in order to demonstrate the effect induced by Palbociclib. Authors should also discuss the difference in total Rb expression.
Response: The reviewer raises a valid point. Though the total Rb band in the palbociclib treated cells does suggest a decrease in total Rb levels, there is in fact a negligible difference in band intensities between vehicle vs palbociclib (33449076 vs 30745771, respectively) compared to that of pRb levels in vehicle vs palbociclib (37760832 vs 17963789, respectively). This has been highlighted in the results section and we have presented the data as % decrease of pRb as suggested by the reviewer (line 273 and figure 1c).
- 1E Are these data not significant? Authors should explain
Response: We apologise for the mistake in this figure, indications of significance levels have now been added (figure 1, page 7).
- 2A There is a discrepancy in the number of T47D cells injected compared to the number mentioned in the material and methods section (5X10^4cells). Indeed, in the figure it is reported 1x10^6 cells. Authors should standardize this data.
Response: We apologise for this error, this has been corrected to state that 1x10^6 cells were injected (line 232).
- The addition of estradiol could affect the primary tumor growth and bone metastasis in the model. Authors should explain whether this compound could affect the results obtained and the Palbociclib effect.
Response: The reviewer is correct, in fact estradiol is essential for ER+ tumour growth as demonstrated by us for the T47D model (Holen et al Clin Exp Metastasis. 2016 Mar;33(3):211-24. doi: 10.1007/s10585-015-9770-x). A reference has been included in the materials and methods section (line 230) of to this study. All animals in the ER+ve study received estradiol (i.e. palbcociclib effects were compared to a control group also receiving estradiol), hence there is no indication that this supplementation affects the reduction in tumour growth or bone metastasis.
- 2C Authors report the number of skeletal tumors. However, this data is not clear, indeed it doesn’t seems authors performed x-rays analysis or µCT to evaluate any skeletal metastasis, which is a consequence of bone homeostasis imbalance induced by tumor cells and the establishment of osteolytic area (in the case of breast cancer). For this reason, authors cannot be sure that cells establish in this model bone metastasis and the data obtained by bioluminescent analysis can only be referred to cell growth capability in the bone. In order to demostrate that cells are able to establish bone metastasis, authors should perform at least x-rays analysis to evaluate any osteolytic area and histology and H&E staining of the bone samples could add information about tumor cells and how they affect bone microenvironment (authors could refer to this paper: doi: 10.1177/0300985815586223 and to this paper https://doi.org/10.1038/s41556-021-00641-w in order to check how to report in the figure the bone metastasis establishment in in vivo experiments). These should be done for all the in vivo experiments in order to demonstrate that cells establish bone metastasis and that Palbociclib can affect this process. Otherwise, it seems that the drug can affect only cell growth and skeletal tumor burden.
Response: We respectfully disagree with the reviewer that uCT analyses are required in order for us to claim that the tumours we observe are indeed bone metastases. Our previous studies and those widely published in the field have clearly shown that both the intra-cardiac TNBC and the ER+ve model is associated with bone destruction, and that the IC injection model is not associated with visceral metastases. Also note that the ex vivo imaging of carefully stripped bones (figure 4D and Supplementary fig 1) demonstrate that tumour growth is in bone as expected in both models. Importantly, the emphasis of our study was not to assess the impact of palbociclib on cancer-induced bone disease, but its ability to reduce tumour growth in bone, as the reviewer correctly states. Throughout our manuscript we refer to tumour growth in bone/tumour burden to describe what we have measured. To carry out the analyses suggested by the reviewer would mean repeating the entire study as the bone samples are not available. The effects of palbociclib on tumour-associated bone destruction is a separate but interesting question that requires further investigation in future studies.
- Authors should explain better the rationale of the different schedule treatment of Palbociclib.
Response: A description of the treatment schedule has been added in the materials and methods section (line 194-198).
- In the introduction, authors should better describe the use of bone targeted agents in bone metastatic patients. Later, they should better explain the rationale for the experiment performed to evaluate the combination of Palbociclib and Zoledronic Acid.
Response: Bisphosphonates were briefly mentioned in line 56. Additionally, a description of the use of zol in bone metastatic patients has already been described in line 421-423. This provides a sufficient foundation for the rationale of the use of Zol in our studies.
- 7 This figure needs a higher resolution because it is difficult to read
Response: Full resolution of FACS plots is added to improve resolution.
- 8 This figure needs statistical analysis.
Response: Statistical analysis has been added to the figure
Reviewer 3 Report
CDK4/6 inhibitors are standard-of-care front line therapy for ER+ metastatic breast cancer but do not have proven benefit in TNBCs. In this study, Saleh and colleagues examine effects of one CDK4/6 inhibitor palbociclib in an intracardiac TNBC model of bone metastasis. Using various dosing schedules, they show that palbociclib monotherapy inhibits BC skeletal tumor burden for a short while but that bone metastases subsequently emerge, that are resistant to palbociclib rechallenge, zolendronic acid, and CDK7 inhibitor. Limited characterization of resistant cells suggests differences in MAPK signaling between palbo-sensitive and resistant tumors. Conceptually, these data highlight challenges of palbociclib for TNBC bone metastasis prevention.
Clinically, the utility of CDK4/6 inhibitors in ER+ breast cancer and in patients with ER+ bone only metastasis is already established, e.g. Rehman et al. Breast Cancer Research and Treatment 2019 among many others for discussion of benefit in bone-met positive population. I do not think there is novelty in notion that CDK4/6 inhibitors are helpful in ER+ bone metastasis treatment, and this study which really focuses on a TNBC dissemination model cannot really make comments about prevention of spontaneous bone metastasis in ER+ breast cancer. In this regard, this manuscript tries to claim too much about treatment of CDK4/6 benefit in bone, and would be stronger narrowing its scientific conclusions to the specific data in hand (TNBC model). While the development of palbociclib-resistant bone metastases is interesting, the features of these cells on a molecular and cellular level are not investigated and should be better characterized.
Major comments:
1. This study hinges on the use of a subline of MDA-MB-231 (ref 30) Nutter et al, which has tendency to home to bone by intra-cardiac delivery. Because many homing lines do not in fact home exclusively to one location, and because this is an intracardiac line which can seed to many organs (brain, lung, liver), the authors should directly confirm by BLI on necropsy and immunofluorescence on tissue sections that their model does not in fact go to these different sites. This seems very important because the authors use skeletal tumor burden (2c, 3c, 4c, 5c, 6c, 7c)- are they just assuming all signal is bone? (e.g. 4d week 8 shows spots that could be lung and brain). In addition to providing more detail in methods, they should also report non-skeletal tumor burden if they are gating the data in some way.
2. The authors place a lot of emphasis on dosing schedule but this is not particularly meaningful clinically. Palbociclib is given continuously once a day 21 days on 7 days off (with sometimes longer breaks) in humans. Why 5 days on 2 days off is more realistic is unclear. Further, abemaciclib- which is considered to be more potent than palbociclib- is given twice daily with no breaks. So a no break CDK4/6 inhibitor is already being used in clinical practice and commonly. The limitations of their conclusions on clinical applicability should be discussed.
3. The emphasis on dosing schedule detracts from the core and potentially interesting observation in this study originating in Fig. 4c, which is that palbocilcib suppresses bone metastases but that this treatment fails somewhat miserably after 4 weeks, off treatment, and even after 1 week off therapy. However, the authors provide only a very limited examination of the resistant tumor cells that arise in this model. Three experiments are suggested:
a. The authors should at a minimum isolate palbociclib-resistant tumor cells and generate sublines to determine if these cells are now intrinsically resistant to palbociclib. One would expect that the IC50 should change compared with Fig.1- alternatively no difference suggests a microenvironmental effect.
b. The authors should dissect bones from week 4 of palbociclib-treated and vehicle-treated animals and demonstrate microscopic metastases at single cell resolution (confocal imaging/IF, or IHC). What is pRB state of resistant cells? This seems very feasible because tumor cells are GFP+.
c. The presumption is that palbociclib suppresses metastases, but an interesting possibility is that palbociclib is selecting for a more aggressive variant. If the authors take cells from experiment A and reinject into animals, are these cells more or less metastatic than parental lines? This seems possible based on BLI in 3b and 4c, that tropism changes in the palbociclib-resistant mets.
Minor comments:
1. Supplementary figures were not available. However, the relative IC50s of the MDA-MB-231 line vs. ER+ cell lines should all be presented for direct comparison of relative sensitivity to palbociclib.
2. Fig. 8 heatmap not interpretable. heatmap scale bar completely white.
3. Tumor burden graphs would be easier to interpret with color figures, not easy to read in their current form.
4. A discussion of potential off-target effects of palbociclib and CDK7 should be provided.
Author Response
- This study hinges on the use of a subline of MDA-MB-231 (ref 30) Nutter et al, which has tendency to home to bone by intra-cardiac delivery. Because many homing lines do not in fact home exclusively to one location, and because this is an intracardiac line which can seed to many organs (brain, lung, liver), the authors should directly confirm by BLI on necropsy and immunofluorescence on tissue sections that their model does not in fact go to these different sites. This seems very important because the authors use skeletal tumor burden (2c, 3c, 4c, 5c, 6c, 7c)- are they just assuming all signal is bone? (e.g. 4d week 8 shows spots that could be lung and brain). In addition to providing more detail in methods, they should also report non-skeletal tumor burden if they are gating the data in some way.
Response: The reviewer raises an important point. We conducted whole body imaging of all tumour-bearing mice to ensure that metastases were located at skeletal sites. In our hands, there are no visceral metastases following intra-cardiac injection of the TNBC cells used in this study; we only observe lung tumours in ‘misplaced’ injections that are evident by a Luc2 signal in the chest a few days following tumour cell injection. In the few cases where this happened, animals were removed from the study and formed no part of the analyses. We therefore do not report any detection of Luc2+ tumour cells in soft tissue and hence our data are from analysis of tumour growth in bone only.
- The authors place a lot of emphasis on dosing schedule but this is not particularly meaningful clinically. Palbociclib is given continuously once a day 21 days on 7 days off (with sometimes longer breaks) in humans. Why 5 days on 2 days off is more realistic is unclear. Further, abemaciclib- which is considered to be more potent than palbociclib- is given twice daily with no breaks. So a no break CDK4/6 inhibitor is already being used in clinical practice and commonly. The limitations of their conclusions on clinical applicability should be discussed.
Response: As our study was the first to investigate the effect of palbociclib in these models, a cautious approach to dosing frequency (5 days on-2 days off) was initially taken to ensure that the drug was tolerated over the 4-week experimental period. Once we had established tolerability, and in light of the relative short half-life of palbociclib in mice compared to humans (https://doi.org/10.1038/bjc.2011.177 and https://doi.org/10.1124/jpet.115.228213) we removed the 2-day break in treatment in order to determine if continuous treatment was effective. Though the reviewer is correct that “a no break CDK4/6 inhibitor is already being used in clinical practice”, this is not the case for palbociclib. Furthermore, CDK 4/6 inhibitors are commonly used in conjunction with endocrine therapy. Here we emphasise the novel finding that palbociclib alone is able to inhibit overt tumour growth in bone in the ER+ and TNBC settings.
- The emphasis on dosing schedule detracts from the core and potentially interesting observation in this study originating in Fig. 4c, which is that palbocilcib suppresses bone metastases but that this treatment fails somewhat miserably after 4 weeks, off treatment, and even after 1 week off therapy.
Response: The reviewer appears to have misinterpreted figure 4c, which shows that tumour growth in bone is completely eliminated after 4 weeks of palbociclib treatment. It is therefore entirely incorrect to say that this treatment ‘fails somewhat miserably’ after 4 weeks, in fact continuous palbociclib treatment is highly effective. However, once treatment is stopped, tumours grow quickly. This is not unexpected, as palbociclib is a cytostatic rather than a cytotoxic drug, with tumour cells able to re-enter the cell cycle once the drug is removed. In addition, we know that in clinical use palbociclib schedules include a treatment break to allow for hematopoietic recovery, hence our exploration of re-treatment schedules and combination therapies.
- However, the authors provide only a very limited examination of the resistant tumor cells that arise in this model. Three experiments are suggested:
Response: We thank the reviewer for these interesting approaches (see responses to each one below). Importantly and as emphasised throughout our manuscript, we do see evidence that the tumour cells become resistant to palbociclib (no tumour growth detected during palbociclib treatment), instead they rapidly lose sensitivity in response to a break in treatment.
- The authors should at a minimum isolate palbociclib-resistant tumor cells and generate sublines to determine if these cells are now intrinsically resistant to palbociclib. One would expect that the IC50 should change compared with Fig.1- alternatively no difference suggests a microenvironmental effect.
Response: The conundrum in our studies is that tumours do not become resistant to the drug; as long as palbociclib is present, tumour growth is effectively suppressed. We have not seen any evidence of outgrowth of resistant clones in any of our many in vivo studies and hence we cannot isolate them as suggested by the reviewer. We did attempt to mimic our in vivo finding by introducing treatment breaks to TNBC cells exposed to palbociclib in vitro, however we were unable to reproduce the re-growth of cells following removal of the drug seen in vivo. This indicates that the reviewers’ suggestion that a microenvironmental element is involved in the in vivo response to a break in palbociclib treatment may be correct and will be explored in future studies.
- The authors should dissect bones from week 4 of palbociclib-treated and vehicle-treated animals and demonstrate microscopic metastases at single cell resolution (confocal imaging/IF, or IHC). What is pRB state of resistant cells? This seems very feasible because tumor cells are GFP+.
Response: This is an interesting suggestion but unfortunately, we do not have the capability to identify single cells in the way that is suggested. We will certainly consider including this in our future attempts to identify and characterise tumour cells in bone that are sensitive to palbociclib compared to those that are not.
- The presumption is that palbociclib suppresses metastases, but an interesting possibility is that palbociclib is selecting for a more aggressive variant. If the authors take cells from experiment A and reinject into animals, are these cells more or less metastatic than parental lines? This seems possible based on BLI in 3b and 4c, that tropism changes in the palbociclib-resistant mets.
Response: We thank the reviewer for this suggestion, however as the parental cell line used in our study generates bone metastases in around 90% of animals it would be difficult to increase this further (assuming selection of a more aggressive variant). A dilution experiment may be an option, but this would be outside the scope of the current study.
Minor comments:
- Supplementary figures were not available. However, the relative IC50s of the MDA-MB-231 line vs. ER+ cell lines should all be presented for direct comparison of relative sensitivity to palbociclib.
Response: The supplementary data had been submitted as per the requirements. Figure 1 is proof of concept that palbociclib inhibits the cell line that we intended to use in vivo. More comprehensive studies comparing IC50 values have been previously published (https://doi.org/10.1186/bcr2419). As requested by the reviewer we have now included the IC50 value of the MDA-MB-231Luc2+GFP+ cell line to the supplementary figure 2.
- 8 heatmap not interpretable. heatmap scale bar completely white.
Response: Again, we believe this is a formatting issue, however we have changed the colour scheme.
- Tumor burden graphs would be easier to interpret with color figures, not easy to read in their current form.
Response: Colour has been be added to the figures for visual aid.
- A discussion of potential off-target effects of palbociclib and CDK7 should be provided.
Response: We did not detect any adverse effects at the 100mg/kg dose in our in vivo studies, even after 4 weeks of daily treatment. This is also the case for the CDK7 inhibitor, hence our data do not suggest that there will be any major adverse events from these treatments. This point is now highlighted in the manuscript. (line 429 and briefly mentioned in lines 665-667).
Round 2
Reviewer 2 Report
Authors have accomplished most of the commests arised in the previous revision. Therefore, the manuscript has been ameliorate so that it can be considered for publication in the journal.
Author Response
We thank the reviewer for their valuable input for the improvement of our manuscript.
Reviewer 3 Report
A few textual clarifications are suggested. First, more detail should be added describing how they performed their intracardiac experiments. As outlined in Response 1, they are excluding some 'misplaced injection' animals, and it would be important to note this step in the methods for reproducibility. Second, the authors should acknowledge the differences between mouse and human CDK4/6 dosing schedules somewhere in the discussion. Third, the authors should consider different explanations for their observations showing that tumors clearly resist palbociclib treatment following 1 week holiday (fig 5). The authors state “the lack of response to palbociclib does not fit with development of resistance, as tumour growth was always strongly inhibited when palbociclib was given daily for up to 4 weeks." But their data indicates that tumor cells after the 1 week holiday are resistant to further palbociclib- which is exactly resistance. Resistance can be innate/from the start or adaptive- in this case adaptive. Whether this adaptive resistance is tumor intrinsic/heritable/reproduced ex-vivo, or tumor extrinsic/a function of the microenvironment is not addressed in this study.
Author Response
- First, more detail should be added describing how they performed their intracardiac experiments. As outlined in Response 1, they are excluding some 'misplaced injection' animals, and it would be important to note this step in the methods for reproducibility.
Response:
Intracardiac tumour cell injections were carried out as described in Wright LE, Ottewell PD, Rucci N, Peyruchaud O, Pagnotti GM, Chiechi A, Buijs JT, Sterling JA. Murine models of breast cancer bone metastasis. Bonekey Rep. 2016 May 11;5:804. doi: 10.1038/bonekey.2016.31. Reference to this detailed description of the procedure is now included in the materials and methos section line 190-191. Futher description of the exclusion of unsuccessfully injected mice has been included in line 194-199.
- Second, the authors should acknowledge the differences between mouse and human CDK4/6 dosing schedules somewhere in the discussion.
Response: The differences between the two schedules have been acknowledged and discussed on lines 678-682
- Third, the authors should consider different explanations for their observations showing that tumors clearly resist palbociclib treatment following 1 week holiday (fig 5). The authors state “the lack of response to palbociclib does not fit with development of resistance, as tumour growth was always strongly inhibited when palbociclib was given daily for up to 4 weeks." But their data indicates that tumor cells after the 1 week holiday are resistant to further palbociclib- which is exactly resistance. Resistance can be innate/from the start or adaptive- in this case adaptive. Whether this adaptive resistance is tumor intrinsic/heritable/reproduced ex-vivo, or tumor extrinsic/a function of the microenvironment is not addressed in this study.
Response: We agree with the reviewer that there could be a number of mechanisms whereby tumour cells become resistant to treatment. We discuss the behaviour of the cells in response to palbociclib (tumour growth inhibition) throughout the manuscript and the surrounding microenvironment (766-770), we now also address the reviewers comment in line 772-774. We agree with the reviewer that different mechanisms for the lack of effect of palbociclib following a treatment break should be considered. However, as explained in the discussion section, we would expect that if it was a matter of outgrowth of palbociclib-insensitive clones (intrinsic resistance mechanism), then addition of a CDK7i in a second treatment cycle should still impact tumour growth. Finally we highlight ways in which further studies may be carried out to identify the mechanisms underpinning the lack of effect of palbociclib (alone or in combination with CDK7i) following a treatment break.